# Lipoprotein(a) and Cardiovascular Outcomes after Revascularization of Carotid and Lower Limbs Arteries

**DOI:** 10.3390/biom11020257

**Published:** 2021-02-10

**Authors:** Marat V. Ezhov, Narek A. Tmoyan, Olga I. Afanasieva, Marina I. Afanasieva, Sergei N. Pokrovsky

**Affiliations:** 1Laboratory of Lipid Disorders, Department of Atherosclerosis, A.L. Myasnikov Institute of Clinical Cardiology, National Medical Research Center of Cardiology, Ministry of Health of the Russian Federation, 121552 Moscow, Russia; 2Laboratory of Atherosclerosis, Institute of Experimental Cardiology, National Medical Research Center of Cardiology, Ministry of Health of the Russian Federation, 121552 Moscow, Russia; afanasieva.cardio@yandex.ru (O.I.A.); miafanasieva.cardio@yandex.ru (M.I.A.); dr.pokrovsky@mail.ru (S.N.P.)

**Keywords:** lipoprotein(a), cardiovascular disease, peripheral artery disease, atherosclerosis, carotid artery disease, carotid atherosclerosis, cardiovascular events

## Abstract

Background: Despite high-intensity lipid-lowering therapy, there is a residual risk of cardiovascular events that could be associated with lipoprotein(a) (Lp(a)). It has been shown that there is an association between elevated Lp(a) level and cardiovascular outcomes in patients with coronary heart disease. Data about the role of Lp(a) in the development of cardiovascular events after peripheral revascularization are scarce. Purpose: To evaluate the relationship of Lp(a) level with cardiovascular outcomes after revascularization of carotid and lower limbs arteries. Methods: The study included 258 patients (209 men, mean age 67 years) with severe carotid and/or lower extremity artery disease, who underwent successful elective peripheral revascularization. The primary endpoint was the composite of nonfatal myocardial infarction, nonfatal stroke, or cardiovascular death. The secondary endpoint was the composite of primary endpoint and repeated revascularization. Results: For 36-month follow-up, 29 (11%) primary and 128 (50%) secondary endpoints were registered. There was a greater risk of primary (21 (8%) vs. 8 (3%); hazard ratio (HR), 3.0; 95% confidence interval (CI) 1.5–6.3; *p* < 0.01) and secondary endpoints (83 (32%) vs. 45 (17%), HR, 2.8; 95% CI 2.0–4.0; *p* < 0.01) in patients with elevated Lp(a) level (≥30 mg/dL) compared to patients with Lp(a) < 30 mg/dL. Multivariable-adjusted Cox regression analysis revealed that Lp(a) was independently associated with the incidence of cardiovascular outcomes. Conclusions: Patients with peripheral artery diseases have a high risk of cardiovascular events. Lp(a) level above 30 mg/dL is significantly and independently associated with cardiovascular events during 3-year follow-up after revascularization of carotid and lower limbs arteries.

## 1. Introduction

Cardiovascular disease (CVD) remains the leading cause of death worldwide. Low-density lipoprotein cholesterol (LDL-C) is the main causal risk factor for atherosclerotic CVD. Recent landmark randomized trials have shown a decrease in cardiovascular events (CVE) with high-intensity lipid-lowering therapy and substantial reduction of LDL-C level [1,2]. However, despite the optimal medical therapy with lipid-lowering drugs, the treatment of diabetes mellitus and arterial hypertension, as well as the use of antiplatelet therapy, the risk of CVD remains quite high. Thus, a significant residual risk remains, despite the achievement of lower and even target LDL-C levels according to the latest recommendations of the European Societies of Cardiology and Atherosclerosis (ESC/EAS) for the management of patients with dyslipidemia [3]. Residual risk may be due to lipid and non-lipid factors. Lipoprotein(a) (Lp(a)) is a risk factor for atherosclerotic CVD and aortic stenosis and may explain, in part, the residual risk of CVD [4]. Hyperlipoproteinemia(a) is one of the hidden lipid metabolism disorders. In the USA, the number of those with a Lp(a) level of more than 30 mg/dL among 531,144 persons examined in one laboratory reached 35% [5].

The FOURIER (Further Cardiovascular Outcomes Research with PCSK9 Inhibition in subjects with Elevated Risk) study showed that in patients with significant lower extremities arteries atherosclerotic disease (LEAD) within 2 years of follow-up the risk of CVE was approximately two times higher than in patients with myocardial infarction or stroke in the past [6]. In addition, patients with LEAD have a high risk of acute limb ischemia, amputation, and revascularization [7]. It has been shown that the risk of CVE in patients with LEAD is even higher in the presence of ischemic heart disease (IHD) and/or cerebrovascular disease [8].

The aim of our study was to evaluate the relationship between the Lp(a) level and the occurrence of CVE after successful revascularization of peripheral (carotid and lower limb) arteries during a 3-year prospective follow-up.

## 2. Material and Methods

The study included 258 patients (209 men and 49 women, mean age 67 years) who underwent revascularization of the carotid and/or arteries of the lower extremities for symptomatic stenosing atherosclerosis. The exclusion criteria were: acute coronary syndromes, infectious diseases in the previous 3 months; chronic kidney disease of stage IV or V; systemic connective tissue diseases; thyroid dysfunction (thyroid-stimulating hormone is 2 times lower than the lower limit of the norm or 2 times higher than the upper limit of the norm); acute hepatitis, liver cirrhosis; chronic heart failure of III-IV functional class; the use of drugs that affect Lp(a) level (nicotinic acid, inhibitors of proprotein convertase subtilisin/kexin type 9).

The study protocol was approved by the local ethics committee. Written informed consent was obtained from all patients prior to enrollment. The study was carried out in accordance with Good Clinical Practice and the principles of the Declaration of Helsinki.

History and physical examination were taken to identify atherosclerosis risk factors. In all patients, serum levels of total cholesterol (TC), triglycerides (TG), high density lipoprotein cholesterol (HDL-C) were determined by the enzymatic colorimetric method using commercial kits “Biocon” (Germany). LDL-C was calculated by the Friedewald formula: LDL-C = TC–HDL-C − TG/2.2 (mmol/L). In addition, the level of corrected LDL-C was calculated, taking into account cholesterol of Lp(a): LDL-C corrected = LDL-C-0.3 × Lp(a)/38.7 [9]. The concentration of Lp(a) was measured using an enzyme-linked immunosorbent assay using polyclonal antibodies to Lp(a), as previously described [10]. The level of C-reactive protein (CRP) in the blood serum was determined using a highly sensitive ELISA kit (Vector-Best, Russia). The creatinine level was determined using a Biocon analyzer (Germany). Glomerular filtration rate (GFR) was calculated by the formula Chronic Kidney Disease Epidemiology Collaboration (CKD-EPI). Chronic kidney disease was diagnosed with a GFR below 60 mL/min/1.73 m^2^.

CVE were registered during 3 years of follow-up. The primary endpoint was the composite of nonfatal myocardial infarction, nonfatal stroke, or cardiovascular death. The secondary endpoint was the composite of primary endpoint and repeat revascularization.

Statistical analysis of the results was conducted with MedCalc 15.8 software (MedCalc Software Ltd., Ostend, Belgium). Descriptive statistics of continuous quantitative variables after analyzing the normality of distribution are presented in the form of median and the 25th and 75th percentiles, qualitative data in the form of absolute numbers and percentages. The Kolmogorov–Smirnov test was used to determine the normal distribution. Fisher’s exact test was used to compare the frequency parameters between groups. Analytical statistics were performed using the Mann–Whitney test. Differences were considered statistically significant at *p* < 0.05. The threshold value of Lp(a), its sensitivity and specificity were obtained by receiving curves of operational characteristics (ROC analysis). To assess the associations between Lp(a) and past myocardial infarction or stroke, odds ratios (ORs) were calculated with 95% confidence intervals (95% CI). Survival analysis was performed using the Kaplan–Meier method with the calculation of hazard ratio (HR) of CVE and 95% CI. Multivariate analysis was performed using Cox regression. When creating the model, the absence of internal correlations between the estimated parameters was also taken into account.

## 3. Results

General characteristics of examined patients are presented in Table 1, 175 (68%) patients had IHD, 85 (33%) suffered myocardial infarction in the past, 99 (38%) subjects were with a history of percutaneous coronary intervention, 56 (22%) had patients coronary artery bypass grafting.

After revascularization, all patients were prescribed dual antiplatelet therapy with a further transition to monotherapy with acetylsalicylic acid or clopidogrel, and statin therapy. By the end of 36 months of follow-up, 2 (0.8%) patients stopped taking antiplatelet drugs and 21 (8%) patients stopped taking statins.

In 258 patients, 294 operations on the peripheral arteries were performed: 171 revascularizations on carotid arteries, 123 revascularizations on lower limb arteries, revascularization in two vascular beds were in 36 patients. The following types of operations were performed: stenting, bypass grafting, endarterectomy of carotid arteries (Figure 1A); balloon angioplasty, stenting, prosthetics, bypass grafting, endarterectomy of lower limb arteries (Figure 1B).

During 3 years of follow-up, the primary endpoint was recorded in 29 patients, including 19 (7%) myocardial infarctions, 15 (6%) strokes, and 6 (2%) cardiovascular deaths. The secondary endpoints were recorded in 128 patients, including repeated revascularization in 111 cases and the primary endpoints in 29 cases. Patients with outcomes had higher incidence of stroke in the past and higher levels of total cholesterol, Lp(a), LDL-C and corrected LDL-C as well as other clinical and laboratory variables did not differ regarding the CVE.

According to ROC-analysis, the concentration of Lp(a) more than 30 mg/dL was associated with CVE after revascularization of peripheral arteries with a sensitivity of 63% and a specificity of 71% (Figure 2). Lp(a) ≥ 30 mg/dL (hyperlipoproteinemia(a)) was not associated with past myocardial infarction and ischemic stroke: OR 1.0, 95% CI 0.6–1.7, *p* = 1.0 and OR 1.0, 95% CI 0.5–1.8, *p* = 0.9, respectively.

In patients with hyperlipoproteinemia(a), more CVE were recorded: 21 primary endpoints and 83 secondary endpoints, while in patients with a concentration of Lp(a) < 30 mg/dL, 8 primary and 45 secondary endpoints were diagnosed (*p* < 0.01 for both). In Kaplan–Meier analysis, the survival curves diverged significantly depending on the Lp(a) level (Figure 3). Hyperlipoproteinemia(a) was associated with an increased risk of primary and secondary endpoints after peripheral revascularization: hazard ratio (HR) 3.0; 95% CI 1.5–6.3; *p* < 0.01 and HR 2.8; 95% CI 2.0–4.0; *p* < 0.01, respectively. During 3 years follow-up repeated revascularization at target vessel due to restenosis was registered in 111 patients. Hyperlipoproteinemia(a) was associated with an increased risk of repeated revascularization: HR 2.9; 95% CI 2.0–4.2, *p* < 0.01.

According to Cox regression analysis with the inclusion of all risk factors in the model, hyperlipoproteinemia(a) remained an independent predictor of CVE and was associated with a three-fold increase in the risk of primary and secondary endpoints (Table 2).

## 4. Discussion

Most studies have shown a relationship between Lp(a) and the development of CVE in patients with IHD. In a Japanese study (*n* = 927, 81% men, mean age 67 years), it was shown that in patients with IHD and type 2 diabetes who underwent percutaneous coronary intervention, the concentration of Lp(a) more than 19.5 mg/dL was associated with a two-fold increased risk of such events as non-fatal myocardial infarction, non-fatal stroke, death from any cause (hazard ratio 1.91; 95% CI 1.20–3.09; *p* < 0.01) with a median follow-up of 5 years [11]. In a study of 356 patients followed during 15 years after successful coronary artery bypass grafting, Lp(a) ≥ 30 mg/dL was associated with a three-fold increase in CVE risk (nonfatal myocardial infarction, unstable angina pectoris requiring hospitalization, repeated revascularization, cardiovascular death) [12]. According to the ESC/EAS 2019 guidelines for the treatment of patients with dyslipidemia Lp(a) level below 50 mg/dL is considered as normal [3]. Recent Canadian guidelines and the American Consensus Panel recommend that an Lp(a) concentration of as low as 30 mg/dL be considered as a cut-off level [13,14].

We have shown that Lp(a) levels greater than 30 mg/dL are associated with a threefold increase in the risk of CVE after revascularization of the lower extremities and carotid arteries over 3 years. According to Cox proportional-hazards model hyperlipoproteinemia(a) has been shown an independent predictor of cardiovascular events adjusted for all risk factors, including past myocardial infarction and stroke. Our data are comparable with the results of a study conducted in Japan involving 189 patients (mean age 72 years, 160 men) with peripheral atherosclerosis who underwent endovascular interventions on the arteries of the aortoiliac segment [15]. Within 3 years, amputation and repeated revascularization were recorded in 44 (23%) patients. These outcomes in the group of patients with Lp(a) > 40 mg/dL were observed more often than in the group with Lp(a) ≤ 40 mg/dL: 45 and 15%, respectively, *p* < 0.001. The independent predictors of events were an increased Lp(a) level (RR 2.8, 95% CI 1.4–5.5, *p* = 0.003) and programmed hemodialysis (RR 2.23, 95% CI 1.04–4.78, *p* = 0.04) [15]. In an Australian study, involving 1472 patients with LEAD, it was shown that the level of Lp(a) ≥ 30 mg/dL was associated with the frequency of peripheral revascularization; however, no association was found between elevated Lp(a) level and the frequency of myocardial infarction, stroke, and death [16]. In our study, there was not any association found between Lp(a) and past myocardial infarction or ischemic stroke, which is most likely due to the fact that the studied cohort of patients with significant atherosclerosis of peripheral arteries has an initially increased level of Lp(a). The frequency of myocardial infarction and stroke in the studied patients is higher than in the general population [17].

Large studies confirm that Lp(a) is a risk factor for CVE, even in patients with LDL-C levels < 70 mg/dL (1.8 mmol/L). In the AIM-HIGH (Atherothrombosis Intervention in Metabolic Syndrome with Low HDL/High Triglyceride and Impact on Global Health Outcomes) study it was shown that in patients with a mean LDL-C level of 65.2 mg/dL (1.7 mmol/L) and Lp(a) ≥ 50 mg/dL, the risk of death, myocardial infarction, and revascularization was 89% higher compared to patients with comparable levels of LDL-C and Lp(a) < 50 mg/dL [18]. In the JUPITER (Justification for the Use of Statins in Prevention: an Intervention Trial Evaluating Rosuvastatin) study patients with LDL-C level 55 mg/dL (1.4 mmol/L), and Lp(a) concentration ≥ 21 mg/dL had a 71% higher risk of CVE compared to patients with Lp(a) < 21 mg/dL [19]. The LIPID (Long-Term Intervention with Pravastatin in Ischaemic Disease) study showed that in patients with LDL-C levels of 112 mg/dL (2.9 mmol/L) and Lp(a) ≥ 74 mg/dL, the risk of CVE was 23% higher compared with patients with Lp(a) < 74 mg/dL and comparable LDL-C [20]. We have shown that patients after peripheral revascularization with and without CVE were comparable by level of corrected LDL-C, but significantly differed by of Lp(a) concentration.

In the FOURIER study, patients were divided into groups depending on achieved LDL-C level: less than 0.5 mmol/L (*n* = 2669), from 0.5 to <1.3 mmol/L (*n* = 8003), from 1.3 to <1.8 mmol/L (*n* = 3444), 1.8 to <2.6 mmol/L (*n* = 7471) and ≥2.6 mmol/L (*n* = 4395). The incidence of CVE was 10.3, 12.4, 13.6, 13.7, 15.5%, respectively, for 2.6 years of follow-up [21]. Thus, even in individuals with an LDL-C concentration of less than 0.5 mmol/L, the rate of CVE was more than 10%, proving that targeted LDL-C alone does not completely eliminate CVE risk.

The question of a reduction of the frequency of CVE with a decrease in Lp(a) level remains open due to the limited therapeutic potential for this particle. Nicotinic acid, which reduces the concentration of Lp(a) by 30%, is banned in Europe due to the high incidence of serious side effects [22]. Large studies have shown a Lp(a) level reduction up to 30% when using proprotein convertase inhibitors subtilisin/kexin type 9 (PCSK9) [23,24]. In the FOURIER study, the use of evolocumab led to a decrease in Lp(a) concentration by an average of 27%, that was associated with a 23% reduction in the risk of coronary events [23]. The ODYSSEY Outcomes (ODYSSEY Outcomes: Evaluation of Cardiovascular Outcomes After an Acute Coronary Syndrome During Treatment With Alirocumab) study showed that alirocumab reduced the concentration of Lp(a) by 23%. It was revealed that a decrease in Lp(a) level by 1 mg/dL was associated with reduction of CVE risk with a HR of 0.994 (95% CI 0.990–0.999; *p* < 0.01) [24]. However, a meta-analysis of 10 ODYSSEY studies showed that a decrease in Lp(a) levels by an average of 25.6% in the group of patients receiving alirocumab did not lead to a significant decrease in CVE regardless of a decrease in LDL-C (HR 0.89; 95% CI 0.79–1.01; *p* = 0.08) [25].

Recently, a drug based on an antisense oligonucleotide has been developed, which directly blocks the synthesis of apolipoprotein(a) and reduces the level of Lp(a) by 80% [26]. Currently, a phase 3 study of pelacarsen is being carried out that will allow to assess the relationship between a sustained and isolated reduction in Lp(a) with a decrease in the risk of CVE.

In several countries, lipoprotein apheresis is used to treat hyperlipoproteinemia(a), especially in combination with familial hypercholesterolemia. In the case of systems for apheresis of atherogenic lipoproteins (LDL-apheresis), Lp(a) is removed along with other classes of apoB-containing lipoproteins. The only study that assessed the clinical effect of the Lp(a) level reduction without affecting other lipid parameters and proving the atherogenicity of Lp(a) was conducted by our group [27,28].

## 5. Conclusions

In a three-year observation study of patients after peripheral revascularization of the carotid and/or arteries of the lower extremities, the Lp(a) level ≥ 30 mg/dL was an independent risk factor for the development of CVE and may explain partly the residual risk.

## Figures and Tables

**Figure 1 biomolecules-11-00257-f001:**
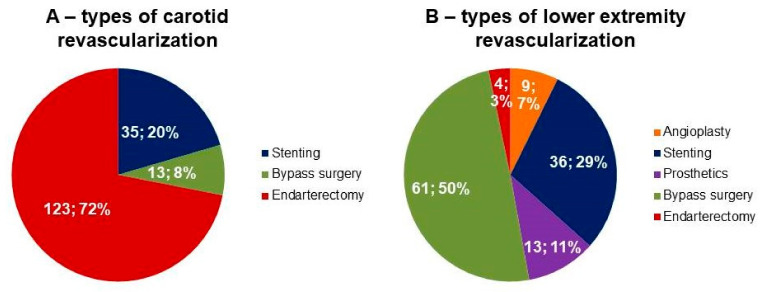
Types of surgical operations. Data are presented as absolute number of patients (%). (**A**) Types of carotid revascularization. (**B**) Types of lower extremity revascularization.

**Figure 2 biomolecules-11-00257-f002:**
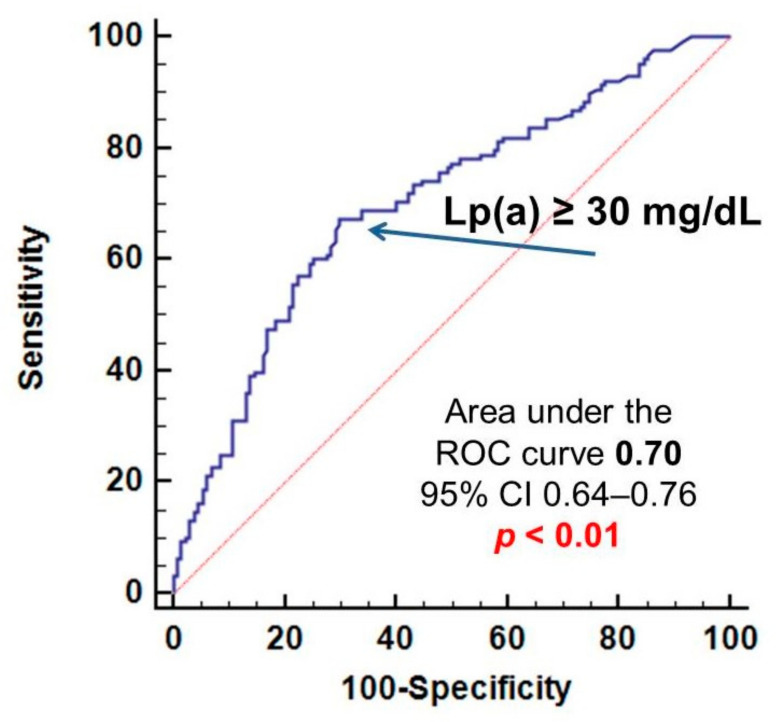
Relationship of lipoprotein(a) with the development of cardiovascular events after peripheral revascularization. Lp(a)—lipoprotein(a), CI—confidence interval. ROC—receiving operating characteristics.

**Figure 3 biomolecules-11-00257-f003:**
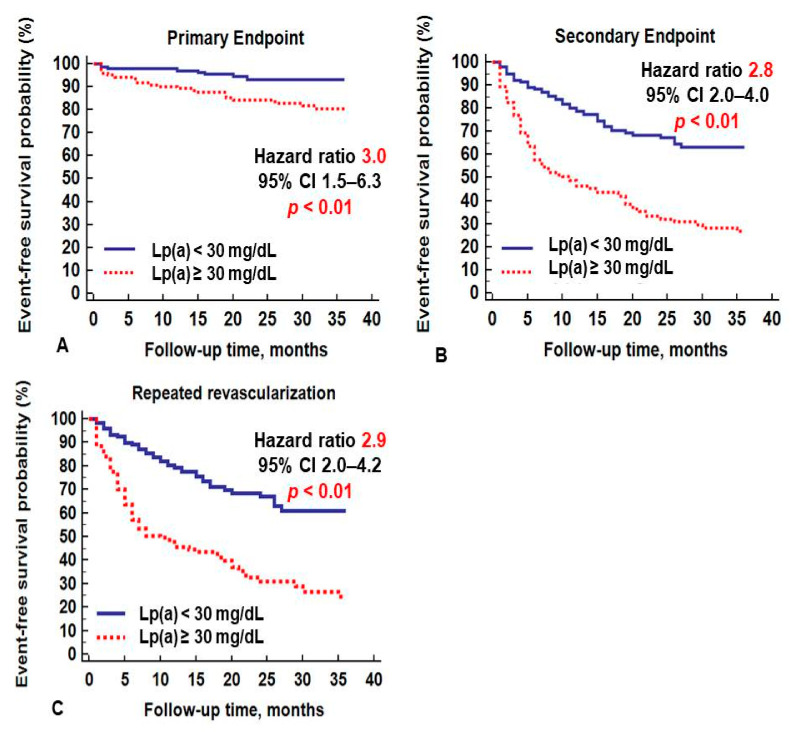
Cardiovascular events depending on the lipoprotein(a) level: (**A**) primary endpoint, (**B**) secondary endpoint, (**C**) repeated revascularization.

**Table 1 biomolecules-11-00257-t001:** Characteristics of patients.

Parameters	Total *n* = 258	CVE+ *n* = 128	CVE− *n* = 130
Age, years	67 [61; 75]	67 ± 8	67 ± 10
Male sex	209 (81%)	107 (84%)	102 (78%)
Hypertension	224 (87%)	108 (84%)	116 (89%)
Obesity	72 (28%)	36 (28%)	36 (28%)
Type 2 diabetes	65 (25%)	38 (30%)	27 (21%)
Smoking	130 (50%)	66 (52%)	64 (49%)
Ischemic heart disease	175 (68%)	93 (73%)	82 (63%)
Myocardial infarction	85 (33%)	49 (38%)	36 (28%)
Ischemic stroke	54 (21%)	36 (28%) *	18 (14%)
Total cholesterol, mmol/L	4.4 [3.8; 5.1]	4.6 [3.9; 5.3] *	4.2 [3.7; 4.9]
Triglycerides, mmol/L	1.5 [1.1; 2.0]	1.6 [1.1; 2.0]	1.4 [1.1; 2.0]
HDL-C, mmol/L	1.2 [1.0; 1.4]	1.2 [1.0; 1.4]	1.3 [1.0; 1.5]
LDL-C, mmol/L	2.4 [1.9; 2.9]	2.6 [2.1; 3.1] *	2.2 [1.8; 2.7]
LDL-C_corrected_, mmol/L	2.0 [1.6; 2.6]	2.1 [1.7; 2.7]	1.9 [1.5; 2.6]
Lipoprotein(a), mg/dL	27 [11; 57]	41 (20; 76) *	18 (8; 34)
C-reactive protein, mg/L	6.8 [2.7; 10.6]	6.7 [2.2; 13.7]	7.3 [3.6; 10.5]
Creatinine, µmol/L	93 [78; 109]	89.0 [75.2; 110.0]	94.0 [80.1; 107.8]

* *p* < 0.01 when compared with the group without cardiovascular events. CVE–cardiovascular events, HDL-cholesterol-high density lipoprotein cholesterol, LDL-cholesterol-low density lipoprotein cholesterol, LDL-C_corrected_-low density lipoprotein cholesterol, corrected for lipoprotein(a)-cholesterol.

**Table 2 biomolecules-11-00257-t002:** Cox proportional-hazards model of risk of development of primary and secondary endpoints.

Parameters	Primary Endpoint				Secondary Endpoint			
	Univariate Analysis		Multivariate Analysis		Univariate Analysis		Multivariate Analysis	
	HR (95% CI)	*p*	RR (95% CI)	*p*	HR (95% CI)	*p*	RR (95% CI)	*p*
Age, years	1.0 (0.96–1.04)	0.96			1.0 (0.98–1.02)	0.9		
Male sex	3.2 (0.77–13.42)	0.06			1.1 (0.71–1.80)	0.6		
Hypertension	0.5 (0.20–1.19)	0.1			0.7 (0.46–1.18)	0.2		
Obesity	1.9 (0.92–3.99)	0.09			1.0 (0.69–1.49)	0.9		
Type 2 diabetes	1.3 (0.60–2.88)	0.5			1.3 (0.88–1.87)	0.2		
Smoking	1.5 (0.69–3.09)	0.3			0.9 (0.67–1.33)	0.7		
Ischemic heart disease	2.7 (0.93–7.61)	0.07			1.2 (0.84–1.83)	0.3		
Myocardial infarction	2.1 (1.26–3.37)	<0.01	2.0 (1.17–3.40)	0.01	1.3 (0.96–1.65)	0.1		
Ischemic stroke	2.6 (1.54–4.31)	<0.01	2.3 (1.38–3.78)	<0.01	1.7 (1.23–2.27)	<0.01	1.7 (1.27–2.35)	<0.01
Total cholesterol	1.0 (0.84–1.30)	0.7			1.1 (0.96–1.16)	0.3		
Triglycerides	0.9 (0.59–1.49)	0.8			1.1 (0.89–1.33)	0.4		
HDL-C	1.0 (0.35–2.89)	1.0			1.0 (0.59–1.57)	0.9		
LDL-C	1.0 (0.91–1.32)	0.3			1.1 (0.97–1.17)	0.2		
LDL-C_corrected_	1.0 (0.79–1.33)	0.9			1.0 (0.88–1.14)	1.0		
Lipoprotein(a) ≥ 30 mg/dL	3.0 (1.35–6.84)	<0.01	2.9 (1.30–6.61)	<0.01	2.9 (1.99–4.12)	<0.01	2.9 (2.03–4.19)	<0.01
Statins	0.5 (0.17–1.42)	0.2			1.0 (0.52–1.77)	0.9		
C-reactive protein	1.0 (0.99–1.02)	0.7			1.0 (0.99–1.01)	0.7		
Creatinine	1.0 (0.99–1.01)	0.6			1.0 (0.99–1.00)	0.7		
CKD	1.2 (0.81–1.79)	0.4			1.0 (0.81–1.22)	0.9		

Note: The multivariate analysis included all factors that showed a significant association with the development of cardiovascular events in the univariate analysis. HR—hazard ratio, CI—confidence interval, HDL-C—high-density lipoprotein cholesterol, LDL-C—low-density lipoprotein cholesterol, LDL-C_corrected_—low-density lipoprotein cholesterol, corrected for lipoprotein(a)-cholesterol, CKD—chronic kidney disease.

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
