# Peer review of "Lipoprotein(a) and Cardiovascular Outcomes after Revascularization of Carotid and Lower Limbs Arteries"

_biomolecules, 2021, doi:10.3390/biom11020257_

Round 1
Reviewer 1 Report
Lp(a) has been reported to be associated not only with cardiovascular outcomes in patients with coronary artery disease, but also as a risk factor for peripheral artery disease in T2DM (Tseng, Diabetic Care 2003) or associated with PAOD (Kosmas, Ann Transl Med 2019; Colledge, J Am Heart Assoc 2020)
In this report, the authors claimed that for patient (N = 258) undertook intervention at carotid artery or at lower extremities, a Lp(a) > 30 mg/dL is associated with cardiovascular events during 3-year follow-up.
In terms of method, the study population is relatively small. To define the primary end-point in such a specific population, restenosis/repeat intervention at target vessel or peripheral arteries will be more relevant than the composite of nonfatal MI, non-fatal stroke, or CV death. The secondary end-point would be the composite of primary end-point and repeat intervention at target vessel.
In the results, those who have CV events on follow-up had higher rate of MI , ischemic stroke in history, higher levels of total cholesterol or LDL, indicating the significance of classical risk factors for CVE. (table 1) Moreover, the history of MI and ischemic stroke, in addition to the high Lp(a), remain to be strong predictors for CVE after intervention to carotid or lower extremity arteries. (table 2) It may be helpful to analyze the association between Lp(a) levels and the back-ground MI or ischemic stroke.
It would be helpful to discuss the possible reasons for the difference in results between this study and that of Colledge, where the Lp(a) was only associated with PAOD but not CV events.
Reviewer 2 Report
This is an interesting paper demonstrating the significance of higher Lp(a) levels (more than 30 mg/dL) for CVE in patients who underwent interventions either at their carotids or their leg arteries.
It has to be taken into consideration that patients who developed new CVE during the 3 years follow-up period had also suffered from a significantly higher number of myocardial infarctions and strokes – in other words, they had a more expressed degree of atherosclerosis. Of course, this may be due to the elevated Lp(a) levels. But the interpretation of the presented data should take into account this fact.
With respect to the 2019 ESC/EAS guidelines the LDL-C concentrations reported in these high-risk patients are clearly higher than recommended. The authors should provide an explanation for this phenomenon.
The paper urgently needs the help of a native English speaker.
Minor comments
Abstract: and repeat revascularization – write “repeated”
Material and methods: the program which was used for statistical analysis is not reported
Page 4 Line 129: and but corrected LDL-C as well as other clinical and laboratory variables – “and but” is not correct
Figure 3: the denomination of the y-axis should read: event-free survival probability
Table 2 clearly shows that anteceding MI and stroke were also significant factors of influence, not only Lp(a) – this should be written in the comment
Page 7 Line 159: In a Japanese study (n = 927, 81% of men, - “of” should be deleted
Page 7 Line 164: In study we performed before with 356 patients underwent coronary artery bypass grafting for chronic IHD – English is not correct
Page 7 Line 169: According to the guidelines of the Canadian Cardiovascular Society for the management of patients with dyslipidemia and American consensus panel that the threshold level of Lp(a) should also be considered as 30 mg/dL – English is not correct
Page 7 Line 173: We have shown that Lp (a) level of more than 30 mg/dL is associated – English is not correct
Page 7 Line 184: So, according to research AIM-HIGH – improve English
Page 7 Line 196: without CVE were comparable be level of corrected LDL-C, but significantly differed by the concentration – “be level” is not correct English
Page 7 Line 204: 0.5 mmol/l – correct would be 0.5 mmol/L (as in other cases)
Page 8 Line 218: alirokumab – alirocumab
Page 8 Line 220: of CVE risl – risk
